# Consumer Distrust about E-numbers: A Qualitative Study among Food Experts

**DOI:** 10.3390/foods8050178

**Published:** 2019-05-27

**Authors:** Annelies van Gunst, Annet J.C. Roodenburg

**Affiliations:** Department of Food, HAS University of Applied Sciences, Onderwijsboulevard 221, 5223 DE’s-Hertogenbosch, The Netherlands; a.roodenburg@has.nl

**Keywords:** E-numbers, food additives, consumer distrust, food experts, healthy product innovation, communication media

## Abstract

Food additives (E-numbers) are allowed in foods, but many consumers have a negative perception of them. The objective was to study the opinion of food experts about the causes and ways to reduce consumer distrust about E-numbers. Thirteen food experts from universities, research institutes, the government, food industry organisations, media, a nutrition information organisation, a consumer association and two other non-governmental organisations (NGOs) were interviewed with a semi-structured topic list, based on a model of risk perception. Interviews were transcribed, coded by an open-coding approach and analysed. Results indicated that, according to food experts, consumer distrust of E-numbers arose from negative communication by traditional media, social media and books. Food experts suggested that the information sources and the reliability of E-number information are important for consumers. Food experts also suggested reducing consumer distrust by avoiding negative label claims and making collective agreements with all parties about honest and transparent communication. According to interviewed food experts, food companies need to explain clearly and honestly why they use E-numbers in food. A nutrition information organisation and the government were often mentioned as appropriate parties to undertake action. The interviews suggested that consumers had no confidence in the food industry.

## 1. Introduction

E-numbers, approved in the EU by European Food Safety Authority (EFSA), are safe, well-examined and evaluated for harmful side effects [1]. However, at the start of this millennium, there was substantial consumer distrust towards the food industry about food safety issues, as illustrated by frequent discussions in various media outlets [2].

Regulation EU 1333/2008 describes a food additive or E-number as “any substance not normally consumed as a food in itself and not normally used as a characteristic ingredient of food, whether or not it has a nutritive value, the intentional addition of which to food for a technological purpose in the manufacture, processing, preparation, treatment, packaging, transport or storage of such food results, or may be reasonably expected to result, in it or its by-products becoming directly or indirectly a component of such foods” [3].

### 1.1. Consumer Perception of E-numbers

Previous studies from various countries have reported that consumers have a negative perception of E-numbers [4,5,6,7]. Haen (2014) called this ‘the paradox of E-numbers’: ‘E’ stands for ‘safe’ as approved by scientific research, but consumers associated them with ‘chemicals in food products with a negative health effect’ [6]. Food additives are considered as unnatural, artificial and unhealthy [4,5,6,7], resulting in consumer purchases of more natural and unprocessed foods without E-numbers [6].

On the one hand, risk analyses of food hazards (Codex Alimentarius Commission) are dominated by science-based decisions on safety and health. On the other hand, the value judgements of consumers include environmental (intensive farming/organic foods), ethical, aesthetic and cultural (cooking and eating culture, taste) aspects. These aspects received little attention [2,6]. Studies of consumer perceptions of genetically modified foods concluded that these value judgements play a significant role in the acceptance of these foods [8]. If consumers see benefits for themselves, the environment and/or the society, they are likely to accept foods or novel technologies. However, when consumers consider these foods/novel technologies as health or environmental risks, perceive no or little benefits from them, or have no knowledge of their risks, they will not easily accept them and may even have a negative perception of them [8,9]. This negative perception is worsened by low levels of trust in the parties responsible for approval of those technologies or food additives [4]. In addition, consumers do not have the knowledge about the exact health impact of E-numbers [10] and cannot estimate the risks independently [4]. Experts considered unbalanced diets and microbiological hazards as the main health risks, but for consumers, food additives and chemical hazards are the main risks [9,11].

### 1.2. Role of the Media in E-number Communication

One of the first signs of distrust of food additives in the media was based on studies conducted by Feingold on additives (colorants) associated with hyperactivity and aggression in children [6,12]. By the mid-1990s, interest in natural foods began to rise at the same time that consumers became more aware of the potential role of additives in foods. However, the assumption that additives were ‘bad’ remained and consumers felt that food additives should be reduced [13]. Consumer demand for information on E-numbers for use in supermarkets led to the publication of a number of E-lists, which were developed in easily understood forms, such as books and smart phone applications, in various countries [14,15,16]. Colour-coded systems for E-numbers were presented, in which red should be avoided, orange should only be used if unavoidable and green should be considered safe. In one of these colour-coded systems, only 22% of all E-numbers approved in EU by EFSA (EFSA) were declared as completely safe [14].

In addition, mass media (including websites, social media platforms, food blogs and mobile applications) played an important role in communicating messages about healthy food and healthy food preparation for the public [2,17,18].

### 1.3. Role of the Food Industry

Food manufacturers have used the fact that artificial additives are not well-accepted by consumers to their advantage to increase sales by reducing additives, replacing artificial additives with additives of natural origin [4,19,20] or replacing the E-numbers in the ingredient lists by their (chemical) additive name. It is noticeable that products today have fewer E-numbers in their ingredient lists compared to the past [21]. This tendency towards reducing E-numbers on food labels and in foods is called Clean label. E-numbers (additives) are important for our food supply to maintain or improve safety and freshness, to improve or maintain nutritional value and to improve taste, texture and appearance [22]. The declaration of E-numbers on food labels is obligatory, but they can be declared either by using their common (chemical) name or their E-number [23].

Clean label is described as “being produced free of ‘chemical’ additives, having easy-to-understand ingredients lists and being produced by use of traditional techniques with limited processing” [24]. So ‘cleaner labelling’, a trend in the food industry, means eliminating ingredients that sound like chemicals or ingredients recognized as artificial. It also refers to production methods that are perceived as less natural [25]. However, so far there is no common definition of clean label. The interpretation of clean label is subjective, making it unsuitable as a claim on food labels [25,26].

Consumers’ negative perception of E-numbers has been studied in the past [4,5,6,7]. However, to our knowledge, the opinion and roles or responsibilities of food experts have not been studied. The health effects of food (including E-numbers) are often in the spotlight. Consumers are informed by the media, the food industry, politics, scientists, consumers organisations and NGOs. Food experts are expected to know the facts and have informed views on the following questions: What do they see as causes of negative consumer perception of E-numbers? And what can they (and their organisations) do to reduce consumer distrust of E-numbers? 

In order to answer these two questions, a qualitative study was performed. The objective was to study the opinion of food experts on the causes and ways to reduce consumer distrust of E-numbers.

## 2. Methods

### 2.1. Design and Theoretical Framework

Due to the dearth of literature on food experts’ views of consumer distrust of E-numbers, a qualitative study, by means of semi-structured individual interviews, was performed between March and June 2013 [27,28]. As a theoretical framework, the risk perception model ‘Factors that influence the risk perception of the consumer’ (Figure 1) was adapted for this study [29]. 

This model describes the objective risk (A, chance and effect) and a description of the risk (B, characteristics of the risk). Consumers have a risk perception, which is not only based on objective information, but also on personal (D) and social/cultural (E) characteristics. This is called characteristics of observation (C, framing). All these factors together determine how consumers perceive a risk (F, E-numbers) and whether they will accept it or not (+/−). 

This model, designed for consumers, was adapted for the study group of food experts. Based on the factors A–F of Figure 1, a topic list for the interviews of experts and their organisations was developed (see Table 1).

In this study, we investigated how the different factors result in consumers’ negative perception of E-numbers. To answer the first research question, ‘what are the causes of the negative consumer perception of E-numbers according to food experts’, the following topics were discussed in the interviews: communication from different organisations about E-numbers (E, Figure 1), availability and reliability of information about E-numbers for consumers (B) and the use of E-numbers/*clean label* in food (A). To answer the second question, ‘what can food experts (and their organisations) do to reduce consumers distrust about E-numbers’, the following topics were discussed in the interviews: ways to reduce the distrust of E-numbers (C), best ways to improve the knowledge, confidence and attitude of consumers towards E-numbers (D) and actions/communication to reduce the distrust of E-numbers (F).

### 2.2. Study Group and Procedure

The HAS University of Applied Sciences has a well-developed network of food experts and food companies. From this network, 14 Dutch food experts with expertise in E-numbers were selected from: universities, research institutes, the government, food industry organisations, NGOs, media (food magazines/blogs), a nutrition information organisation and a consumer association (see Table 2). 

The aim was to have a broad representation of different sectors: public, private, academic and societal. This was important to maximise variation in opinions of food experts for this qualitative study. Table 2 shows the organisations and their target groups for communication. The organisations did not distinguish between specific consumer target groups with respect to their communication. Almost all (13 out of 14) of the contacted experts agreed to participate in this study; one expert could not participate due to time constraints. The selected experts were first approached by phone and email (letter) and, if needed, additional information was provided. All experts gave their informed consent for inclusion in this study before they participated in this study. Shared information could not be traceable to them and it was agreed that the experts spoke as individuals (not on behalf of their respective organisations). 

### 2.3. Data-Analysis

All interviews were recorded and transcribed. They were summarized and a member check was conducted to validate all interview summaries. The open coding approach was used to analyse the data. Based on preliminary examination of all data, a code list was discussed and developed to analyse the interviews. Three different researchers independently coded interviews with the aid of this list, resulting, after discussion, in a clear and coherent content analysis system for all interviews [27,28]. Data were analysed in different phases: open coding (fragmentation), axial coding (coding data with a developed data system (topics and key themes) and selective coding (looking for connections and statements) [27]. In the final phase, the integration and analysis of all data took place. During the process of data collection, the subject of data saturation had been taken into account [30]. All topics in the topic list (Table 1) were covered after 13 interviews and the experts provided sufficient information on all topics. During the coding process of the last interviews, no new codes were added. As such, data saturation was achieved [30].

## 3. Results and Discussion

### 3.1. What Are the Causes of Consumer Distrust of E-numbers?

The most important causes for consumer distrust mentioned in the interviews were: the perceived reliability of the source of E-number information, the domination of negative information on E-numbers and the trend towards natural E-numbers and *clean label* food products. All are discussed below.

#### 3.1.1. The Perceived Reliability of the Source of E-number Information

According to all interviewed food experts, including the interviewed experts from food industry (branch) organisations, the food industry was not seen as reliable to communicate about nutrition and health. The literature confirmed this [7,31,32,33]. However, there are still some opportunities for food companies to play a role, according to the experts in this study:“*There is also a task for the retailers, who all have their own magazines. They can explain (……) what they do with E-numbers in their products. With this they can distinguish themselves with their private label offerings*”.(expert from research institute)

Respondents stated that recently consumers also had little trust in other parties if they had contact with the food industry:“*For many years, the industry has not been seen as a reliable information source on nutrition & health, but more recently consumers also have less confidence in neutral parties when they have contacts with food companies*”.(expert from food industry organisation)

This might be a reflection on the increasing public-private cooperation, which compromises the independence of food experts, making them less trustworthy to consumers [34].

Despite this, the food experts from different organisations agreed that there was sufficient reliable information about E-numbers available for consumers. A nutrition information organisation and a consumer association were generally regarded as reliable sources of information about E-numbers, according to nearly all interviewed experts:“*A consumer association and a nutrition information organisation have good websites for information and Wikipedia is surprisingly neutral on the subject of E-numbers. So there are sufficient sites available with good and objective information*”.(expert from university)

Other consumer studies confirm the reliability of the consumer association and the nutrition information organisation’s websites [31,33,35]. The literature describes indicators of reliability as the absence of a commercial interest and the fact that the sender does not pursue its own interests but a societal interest, such as the environment or animal welfare [31]. If consumers distrust the sender, the risk perception of additives increases and the food acceptability decreases [4].

Other reliable organisations or sources mentioned in this study were: Wikipedia, animal rights organisations (NGOs) and the government. Reliable sources of information on E-numbers that were reported elsewhere included: the government, NGOs [35], scientists from universities and food labels [7].

In summary, according to the food experts a nutrition information organisation, a consumer association and the government were generally regarded as reliable sources of E-number information, the food industry was not seen as a reliable source.

#### 3.1.2. Domination of negative information on E-numbers

Generally, all respondents agreed that information on E-numbers was available on the internet and easy to find, but it was difficult for consumers to determine the information’s reliability (quote 1). Negative information about E-numbers strongly dominated search results (quote 2).

(1) “*With Google you should be very careful, you should look at the source. But the question is whether consumers can critically judge the results of a Google search*”.(expert from research institute)

(2) “*You can Google and find a lot about E-numbers. The negative news on E-numbers strongly dominates, therefore it is logical that people have the impression that there is something not right. It is important to note that also many professionals go along with the negative information on E-numbers*”.(expert from university)

Communication tools offered by the food experts’ own organisations varied, yet most communicated through websites. Generally, E-numbers were just one of the topics covered. According to respondents, the websites of a nutrition information organisation and a consumer association were often consulted by consumers for E-number information. The nutrition information organisation optimized their internet search engine strategy so that when people searched for ‘aspartame’, the organisation’s credible website was listed first, before anti-aspartame websites. 

In this study, less frequently used communication tools were also mentioned, such as reports, fact sheets, face-to-face counselling, newsletters, apps and books. Some tools were only used by expert organisations if consumers/food professionals or the media had questions. In literature, leaflets and pamphlets were also mentioned as useful materials for consumers to understand the various types of food additives [36,37]. The important role of scientific expert organisations to give consumers reliable information, as shown by Van Dillen et al. (2004) [35] in their study of safety and health related subjects by Dutch consumers, was not confirmed in our study. Van Dillen et al. (2004) [35] mentioned information sources, including industry-related product boards (such as Dutch Dairy Centre), written education materials and scientific organisations. An explanation for these differences could be that in the last ten years the role of the media (including websites, Facebook, food blogs and mobile applications) played an increasingly important role in the communication of messages about healthy food and healthy food preparation [17].

Thus according to the food experts, information on E-numbers was available on the internet and easy to find, but it was difficult for consumers to determine the information’s reliability; also the negative information strongly dominated.

#### 3.1.3. Trend Towards Natural E-numbers and Clean Label Food Products

Experts in this study mentioned that consumers think that *natural* is safe. However, this is not always the case. If food companies communicate that their products are reduced in E-numbers, they suggest that these E-numbers are bad for health:“*Consumers have the idea that ‘natural’ is the same as ‘safe’, this is wrong: there are many toxic substances in nature and many E-numbers are natural. Big food companies communicate ‘without this E-number’ and ‘without that E-number’, but they actually communicate that what they have taken out is not good for you*”.(expert from university)

It has been frequently reported that E-numbers are considered by consumers as unnatural, artificial and unhealthy, resulting in the purchase of more natural and unprocessed foods without E-numbers [4,6,7]. Despite the fact that the claim natural is used on food labels, the legislation of additives (E-numbers) does not differentiate between natural and artificial E-numbers [3,4]. Consumers use natural as a simple feature of food products representing superior attributes. They are perceived to be healthier and less harmful than conventional products [38]. However, natural does not imply indisputable safety compared to artificial origin. This information might be important for the consumer.

The reaction of the food industry to consumer demand for fewer E-numbers is to move towards clean label (no common used definition available). Most of the food experts in this study described clean label as follows: ‘As little E-numbers as possible, written in words or completely removed from the product’. The food experts mentioned important consequences of clean label: shorter shelf life, lower quality, and rapid deterioration of colours and flavours (more waste); and with respect to health aspects, fewer E-numbers might mean more sugar, salt and fat. This clean label or natural trend is certainly not new [19,20,24,37]. Mentioned disadvantages of the clean label trend included price (alternative natural ingredients are generally more expensive), shorter shelf life (E- numbers are preservatives) and health risks (replacement of sweeteners by sugar make products less healthful) [39].

The clean label trend aims to avoid E-numbers, which generates a natural perception, by using ingredients that are familiar to consumers [25,40]. The claim natural has become one of the leading label claims on new products both in the EU [41] and the US [42]. But how important are those claims for consumers while shopping? Many studies have looked at consumer use of food labels, but most of these are questionnaire-based and not based on the actual shopping behaviour of consumers. For example, Grunert, Wills, & Fernández-Celemin (2010) stated that only 27% of shoppers reported looking at nutrition information (GDA labels and nutrition table) on labels in the UK [43].

The call for natural E-numbers and clean label foods might distract from real health issues [44,45,46]. Excessive intakes of energy, salt, saturated fat and sugar can lead to an increased risk of chronic diseases, such as cardiovascular diseases, cancer and diabetes [47,48,49,50], therefore many initiatives exist to reduce these nutrients. Some of these nutrients, such as salt and fat, have technological functions. Their removal requires the use of additives (E-numbers) to substitute for their functions. Also, removing sugar from foods requires the use of sweeteners (additives) to maintain a certain taste profile. As such, clean label may distract from what is really important in terms of health: reducing the levels of salt, sugar and saturated fat [44,45,46]. Consumers want to avoid E-numbers but are generally unaware of the consequences of removing them, such as unhealthier products, shorter shelf life and reduced quality. Therefore, it is necessary to communicate these consequences to consumers. 

Recently, a study of the Dutch Food Safety Authority (NVWA) (2018) found that the main health risks expressed by consumers were the amount of sugar and salt in food products, closely followed by pesticides and food additives [33]. This is a good development, because too much sugar and salt are more a concern for public health than E-numbers.

In summary, according to the food experts, consumers ask for natural food products and fewer E-numbers in food products. The reaction of the food industry is to produce clean label food products. This might distract from the intakes of less energy, salt, saturated fat and sugar in foods, which are more important to improve public health.

### 3.2. What Are the Best Ways to Reduce Distrust of E-numbers?

Communication around E-numbers needs to improve, especially considering the domination of negative information in the media and the trends toward clean labels and natural products. The interviewed food experts provided their opinions on ways to communicate (actively or passively), which messages should be communicated, and which actions should be taken, including modifications to food labelling.

#### 3.2.1. Methods of Communication

Experts from the universities mentioned different ways to communicate E-numbers to consumers. It is important to engage with influential people and organisations (quote 1), to take the emotions of the consumers seriously, and not to talk only about the health and safety of E-numbers, but to also find out what is important for consumers (quote 2).

(1) *“It is important to reach as many influential people as possible such as journalists, large food companies and consumer organisations. Those are the people that bring the information on E-numbers to the general public”.*(expert from university)

(2) *“About the communication between technologists and marketers: I would like to create a dialogue. Not on what is good or what is bad, but on how to discuss the E-numbers. This discussion could be improved if technologists are not only concerned about the health and safety of E-numbers, but also, by asking questions, find out what the consumers find important. (…) Recommendation is to take the emotions of consumers serious and talk about them”*. (expert from university)

Many possibilities for reducing distrust around E-numbers were mentioned in the interviews, such as making collective agreements about the declaration of E-numbers and stopping the use of ’negative claims’ on food labels (e.g., no E-numbers or only natural colours or additives). The majority of consumers did not previously worry about E-numbers. Therefore, communication such as ‘do not worry about aspartame’ created the impression that aspartame may be harmful. It is important for food manufacturers to tell their own honest stories about their foods, showing that they are healthful and safe for consumers: “Negative information *should not be used.* Only discuss *it when requested*, but not in an *active way. It is also very important to try* not *to refute everything, but to tell an own story. This can be seen in debates in politics*. *Trying to* refute *something often reinforces it. So it is better to tell the world* an original but *honest story".*(expert from university)

Thus according to the food experts, it is important to make collective agreements about the declaration of E-numbers and to stop negative claims on food labels and in communication.

#### 3.2.2. Communicated Messages

All interviewed food experts said they are as objective as possible in their own communication. They only communicate scientifically-proven information (for example, EFSA reports), and also found that communication about possible side effects of E-numbers would unnecessarily alarm consumers. Because of this, most organisations did not communicate about them at all. A nutrition information organisation only communicated on the negative perception of intense sweeteners when misinformation led to higher consumption of unhealthier sugar-containing products.

*“People prefer sugary drinks because they are afraid of aspartame. On this point our organisation wants to remove the unjustified fear that causes consumers to eat unhealthy. The intention is not to focus on the E-numbers, but to focus on healthy food. This is what we would like to achieve”*.(expert from nutrition information organisation)

This organisation would proactively encourage consumers to drink low-calorie diet drinks, or to drink water, coffee and tea without sugar. Another consideration was that consumers who avoid E-numbers also avoid processed foods and eat more fruits and vegetables. This was seen as positive, and thus it is important to consider how to inform consumers about E-numbers.

Food experts are aware of consumers distrust for E-numbers. They provided different views on the best way to discuss E-numbers with consumers. In general, the interviewed experts responded passively and only when they were asked.

*“If there are individual questions we answer them. But if there are more questions on E-numbers, we offer an explanation on the website, including Twitter. This works well, looking at the responses”*. (expert form a NGO)

An expert from a food industry organisation noted being very careful with responses, due to the fact that the food industry was not well-trusted by consumers. An expert from a food magazine only wrote about important information or news for food companies, such as new research findings. The consumer association had a section ‘facts and myths’ on their website.

In summary, all interviewed food experts said they are as objective as possible in their communication, they communicate carefully and only scientifically-proven information.

### 3.3. Which Actions Have to be Taken By Whom?

A university expert encouraged a discussion between the government, food companies (technologists and marketers) and consumers, asserting that different parties should have greater dialogue. It would be a breakthrough if food companies would start listening to the arguments and emotions of consumers. Both respondents from the university and the research institute (quote 1 and 2) argued that merely confirming the safety of E-numbers will not address consumer. The researchers explained that ‘gut feelings’ of consumers also needed to be addressed. Organisations, such as NGOs, the Slow Food Youth Network and animal rights organisations, can play a role in this communication.

(1) *“Trust is very much dependent upon the fact that people are on speaking terms. Companies should do this if consumers have problems with E–numbers. In this case NGOs are good parties. Personally I think the most important party is the Slow Food Youth Network (SFYN). The SFYN is a movement with young professionals that is committed to a fairer and healthier food system. (……) Within companies, marketers and technologists should also have more discussions, for example on the topic whether health is most important or not”*. (expert form university)

(2) *“To achieve this the manufacturers and retail are not suitable, the government or an organisation as Animal rights or a consumer association or other independent consumer organisations can communicate this”.*(expert from research institute)

Having an action plan for the communication of E-numbers to reinforce consumer trust is important, according to the interviewed experts. The literature confirmed this statement [25,51]. This is especially important if consumers have little information on new technologies or new ingredients. The importance of a trusted sender for risk communication, as stated by experts in this study, was confirmed by others [4,31]. The experts in this study argued that a nutrition information organisation must take the lead in the communication, the government is also often mentioned, together with other parties such as a consumer association.

In a recent Dutch consumer study, the consumer association was most trusted, followed by the Dutch Food Safety Authority and a nutrition information organisation [33]. They should distribute objective information and bring more positive news in the media about the safety and functionality of E-numbers to reduce misinformation from being propagated.

In summary, the experts in this study argued that a nutrition information organisation must take the lead in the communication, the government is also often mentioned, together with other parties such as a consumer association. Having an action plan for the communication of E-numbers to reinforce consumer trust is seen as important.

#### 3.3.1. Information on Food Labels

Various experts in this study suggested that more transparency in notation is the best strategy for improving the perception of E-numbers among consumers. A starting point would be for food companies make collective agreements about the avoidance of negative claims such as no/reduced E-numbers and the notation of E-numbers. However, this could be difficult to implement because, according to this study, many food producers distinguish their products with these negative claims.

#### 3.3.2. The Food Industry

Today, the food industry addresses food additives in terms of safety and health issues, but Hauser describes the need for companies to regain consumer trust by providing solutions that also consider ethical criteria, including authenticity/naturalness, quality and convenience [52]. According to food experts in this study, the food industry should tell an authentic and honest story to make the case that products with E-numbers are not harmful. Food companies should explain which E-numbers are used in their products, as well as their function and their origin. Asioli et al. (2017) described this: “engage in consumer education about certain ingredients that might be misconceived by consumers in a targeted way that corresponds to consumer involvement levels and processing of information” [25]. It is contradictory that a list of E-numbers, meant to inform consumers about safe additives (approved in EU by EFSA), has become a ‘black list’. The food industry, who supported the development of the list, has now undermined it.

Thus, according to food experts in this study, the food industry should tell an authentic and honest story to make the case that products with E-numbers are not harmful.

### 3.4. The Model ‘Factors That Influence the Risk Perception of the Consumer’

This model was used because it identified crucial factors in risk perception. These factors included the following: objective risk, characteristics of the risk, characteristics of observation (framing) with personal and social/cultural characteristics and the appraisal of the threat. For this study the model, which was designed for consumers, was used to identify a topic list suitable for our study group of food experts. Two of the different factors the model used to clarify the negative consumer perception of E-numbers, as seen by food experts, were especially of interest: Fist, factor E (social/cultural characteristics; target groups and the communication to target groups, Figure 1) seemed to be of less importance in this study because experts from the organisations did not report having specific consumer target groups in their E-number communication. This was unexpected because in the media, E-numbers were previously associated with hyperactivity and aggression in children [6,12], which could have implied that, for example, parents should be an important target group for communication. Second, factor B (characteristics of the risk) played a crucial role in this study. This factor was described by Van Kreijl et al. (2006) [29]: more negative than positive information (framing), the risks are unknown (knowledge), consumers are less inclined to trust the information (trust), controllability of the risk is outside the person (manageability), and involuntary exposure to a hazard (voluntary). All these aspects were mentioned in the interviews. This model of risk perception provided added value in this study, as it clarified the different aspects of risk perception of E-numbers and answered both research questions. 

### 3.5. Strengths and Limitations of This Study

This qualitative study describes the opinions of food experts on consumers’ negative perception of E-numbers. Despite the fact that this study was carried out several years ago, results are still valid. To our knowledge, this has not been studied before.

#### 3.5.1. Strengths

The following methodological aspects have been taken into account to guarantee the quality of the input of information about E-numbers: selection of experts, information check by the interviewees and the coding process and data saturation.

First, it was important to receive diverse input to discover differences and similarities in the E-number discussion. Selection of experts from different organisations, including universities, research institutes, the government, food industry organisations, NGOs, media (food magazines/blogs), a nutrition information organisation and a consumer organisation guaranteed this. Almost all (13 out of 14) of the contacted experts of these organisations agreed to participate in this study. Second, we verified that the information from the interviews was well-understood. Therefore, all interviews were summarized and a member check was conducted to validate the summarized interview text. Finally, the coding process itself: the interviews were coded (structured around topics and key themes) by three different researchers and the differences in coding were discussed. It was noticed during the coding of the last interviews that no new codes had to be added to the system, therefore data saturation had been reached.

#### 3.5.2. Limitations

This study was based on 13 interviews with food experts of different organisations in the Netherlands. These experts spoke as individuals and not on behalf of the organisation about what consumers think about E-numbers. Therefore, other experts in the Netherlands or EU could have other opinions or could have mentioned other details. Thus, it must be noted that this study does not necessarily reflect the opinions of all food experts. This study was based on expert findings, so the real causes for consumer distrust may be different than what these experts think.

## 4. Conclusions

In summary, the consumer distrust arose from negative communication in traditional media, social media and books. According to food experts, the sender and the reliability of E-number information are important for consumers. Food experts suggested to reduce distrust by avoiding negative claims on the label, such as no/reduced E-numbers, and to make collective agreements with all parties about honest and transparent communication to consumers. Food companies need to explain why they use E-numbers in food. Use of target group-specific communication should be considered. A nutrition information organisation, a consumer organisation and the government were often mentioned as parties to take the lead. Consumers had no confidence in the food industry, according the experts. This lack of confidence in the food industry can be a topic of further study, as there might be other consequences, besides the negative perception of E-numbers.

### Future Research Directions

Also, the underlying causes and possible solutions to improve the food industry’s reputation could be examined. Furthermore, it would be interesting to study if the negative perception of E-numbers by consumers actually influences their buying behaviour in supermarkets and or prevents consumers from buying healthy food.

## Figures and Tables

**Figure 1 foods-08-00178-f001:**
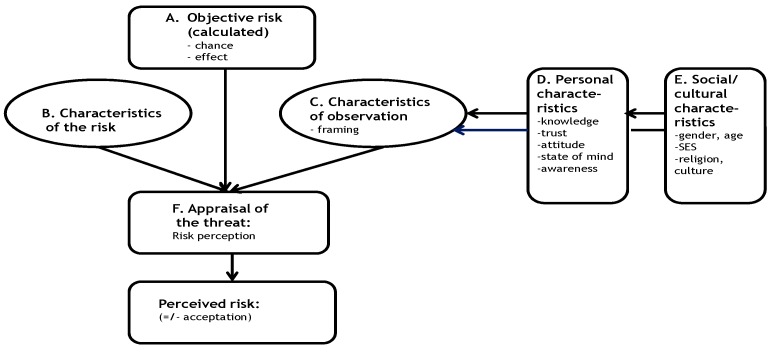
Factors that influence the risk perception of the consumer (adapted from [29]).

**Table 1 foods-08-00178-t001:** Topic list interviews based upon 2 research questions and risk factors of Figure 1 (A t/m F).

Research Questions	Factors (A–F, Figure 1)	Topic List Interviews about E-numbers
**1. What are the causes of consumer distrust of E-numbers?**	E. Social/cultural characteristics	**Communication of different organisations** Intentions in communication (aim, target groups, ways of communication to different target groups).
	B. Characteristics of the risk	**Availability and reliability of information for consumers.** Quantity of information available for consumers, positive/negative information, reliability of information. How do consumers deal with it? Where does the distrust of E- numbers come from?
	A. Objective risk (calculated): change and effect	**The use of E-numbers/clean label in food** Change: Probability that food with E-numbers is not safe and will harm human health. Effect: objective effects, scientific proven knowledge
**2. What are the best ways to reduce the distrust about E-numbers?**	C. Characteristics of observation: framing	**Ways to reduce distrust** How to reduce negative information on E-numbers?
	D. Personal characteristics	**Best ways to improve knowledge, confidence and attitude of consumers** Personal characteristics are dependent of knowledge, confidence, attitude, mood and awareness of consumers. What has already been done to improve this and how can this be done in the future?
	F. Appraisal of the threat: risk perception	**Actions/Communication to reduce the distrust** Which organisations can undertake actions/communication to reduce the distrust of E- numbers?

**Table 2 foods-08-00178-t002:** Study group food experts in the Netherlands: organisations and main target groups for communication (*n* = 13).

	Communication Target Groups				
Study Group	Scientists	Food Professionals	Food Industry	Government	Consumers
University 1 (Researcher)	**X**	**X**			**X**
University 2 (Philosopher)	**X**				**X**
Research institute 1	**X**	**X**	**X**	**X**	**X**
Research institute 2			**X**		
Government		**X**	**X**	**X**	**X**
Food industry organisation 1		**X**	**X**	**X**	
Food industry organisation 2		**X**	**X**	**X**	**X**
NGO 1		**X**	**X**	**X**	
NGO 2			**X**		**X**
Media (Food magazine /blogs) 1					**X**
Media (Food magazine /blogs) 2		**X**	**X**		
Nutrition information organisation		**X**			**X**
Consumers’ association					**X**

X = Main target groups for communication.

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
