# Peer review of "Consumer Distrust about E-numbers: A Qualitative Study among Food Experts"

_foods, 2019, doi:10.3390/foods8050178_

Round 1

Reviewer 1 Report

The manuscript entitled „Consumer distrust about E-numbers: a qualitative study among food experts” is interesting, however requires some some amendments.

Authors should present how their study could fill the scientific "gap" and it should be more  justified.

Abstract:

-        Line 9 – “E-numbers are allowed in foods” it should be “Food additives (E-numbers) are allowed in foods”

-        Lines 17-18 – “The information sources and the reliability of E-number information are important for consumers.” – It does not results from the study. It should be communicated as follows “Food experts suggested that the information sources and the reliability of E-number information are important for consumers.”

-        Line 20 – According to the EU law, food producers are obligated to inform the consumers (clearly and honestly) about the role of the food additive.

Introduction:

-         Line 59 – Please, remove the dot 

Materials and methods:

-        Line 105 – we are 6 years after conducting the study. What do authors think about possible change in this analyzed aspect.  

-        Table 2 – Some typos occur (double brackets)

-        How were experts chosen for the study?  Please specify it.

Results:

-        Line 158 – Please, remove the dot 

-        Lines 243-245 – These sentences are not always true. They should be reformulated and more balanced opinion should be presented.

-        Line 296 – “This is good development. “ This sentence should be broaden.

Author Response

see letter in document

Reviewer 2 Report

As the authors refer to as limitations in the text, I believe that this study used very few samples (only 13). However, it is very meaningful study to grasp consumers' perception from the perspectives of experts.

As the authors acknowldege, there are now many comparisons of consumers' and experts perceptioin of E-numbers in the literatures. Additions to the literature at this stage need to be really tight researchs in order to warrant publication. This study may not meet this bar. Page 2 top, the authors discuss the literature on differences of consumers versus experts perceptions of E-numbers. The authors' research focuses on experts side and this only dealt with one of two things.

The ways to reduce the distrust about E-numbers the experts suggested may not work for real consumers. 

The sample sizes are too small in the survey. An increase in the N's may results in more significant differences. 

From page 6, the authors listed and argued responses from 13 experts. To help the readers better understand, summary results need to be presented.

Author Response

see comment in letter

Round 2

Reviewer 1 Report

The manuscript has been improved according to the comments and suggestions.

The manuscript should be prepared according to the temple (e.g. size of the fonts; reference style, etc.)